

# Atmospheric moisture tracking with WAM2layers v3

Peter Kalverla[1], Imme Benedict[2], Chris Weijenborg[2], and Ruud J. van der Ent[3]

[1]Netherlands eScience Center, Amsterdam, Netherlands
[2]Meteorology and Air Quality Group, Wageningen University and Research, Wageningen, Netherlands
[3]Department of Water Management, Faculty of Civil Engineering and Geosciences, Delft University of Technology, Delft, Netherlands

**Correspondence:** Peter Kalverla (p.kalverla@esciencecenter.nl) or Ruud J. van der Ent (r.j.vanderent@tudelft.nl)

**Abstract.** This manuscript documents the atmospheric moisture tracking model WAM2layers v3 (Water Accounting Model - 2 layers - version 3). WAM2layers may be used to gain understanding of atmospheric dynamics and to study rainfall patterns and extremes by mapping their sources or sinks, often in the context of climate and land-use changes. To this end, WAM2layers solves a prognostic equation for tagged moisture in gridded atmospheric datasets such as reanalysis data or climate model output. WAM2layers can be used in forward mode to determine where evaporated water eventually precipitates, or in backward mode to determine where precipitation originally evaporated.

WAM2layers v3 represents a complete rewrite of the WAM2layers model originally introduced in 2010 and subsequently used in more than sixty academic studies. This latest version incorporates performance optimisations to cope with the increased resolution of input data, and introduces various best practices aimed at improved user-friendliness and software sustainability. Since an increasing number of researchers is using the code, this manuscript is intended as an updated description and reference in the academic literature. After describing the history, model formulation, and numerical implementation, we present and evaluate two example cases to illustrate the use and skill of WAM2layers v3. We then discuss best practices, some important assumptions, and directions for future development.

## 1 Introduction

This paper documents the Water Accounting Model – 2 layers version 3, or WAM2layers v3 for short. WAM2layers is a moisture tracking model that can be used to study the transport of water in the atmosphere, from source (surface evaporation) to sink (precipitation) or vice versa. Since the previous version (see Van der Ent et al., 2014, Appendix B), the model has seen some substantial upgrades, and this manuscript is intended as an updated reference for those who have used, are using, or consider to use WAM2layers in their research. Before we delve into details, we briefly consider the scientific and historical context.

Moisture tracking in general has been applied in different fields (e.g., Gimeno et al., 2020) and research objectives range from obtaining better process understanding of specific events (e.g., Dirmeyer and Brubaker, 1999) and different modes of climate variability (e.g., Miralles et al., 2016) to general aspects of the global hydrological cycle (e.g., Theeuwen et al., 2023).





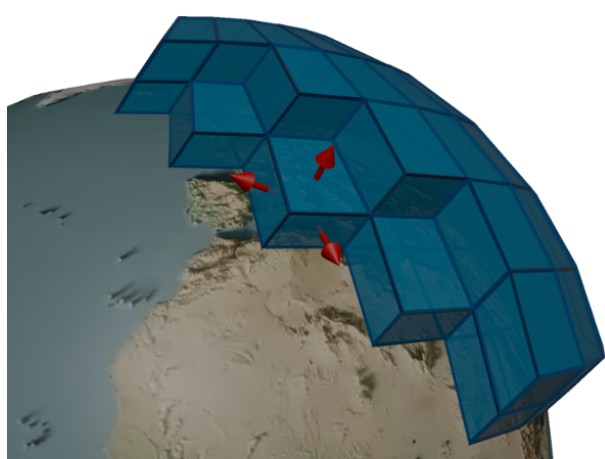

**Figure 1.** Conceptual illustration of WAM2layers. The atmosphere is divided in grid cells, covering the globe partly or fully. There are two vertical layers. Tagged moisture is transported across the interfaces between grid cells.

Moreover, moisture tracking results have often been used in the context of studying the role of vegetation, potential effects of
land-use changes and possible management options (e.g., Te Wierik et al., 2021).

Various types of moisture tracking models exist. A potential classification system was given in Dominguez et al. (2020, Fig. 1), who distinguished between analytical, offline numerical, and online numerical methods. Analytical models are based on budget equations over larger domains and assuming trajectories over such a domain (e.g., Brubaker et al., 1993; Burde and Zangvil, 2001b; Eltahir and Bras, 1994; Savenije, 1995; Schär et al., 1999). Offline numerical models operate on gridded
output fields of weather and climate models or reanalysis data (e.g., Cheng and Lu, 2023; Dey and Döös, 2020; Dirmeyer and Brubaker, 1999; Goessling and Reick, 2013; Gimeno et al., 2020; Holgate et al., 2020; Keune et al., 2022; Sodemann et al., 2008; Tuinenburg and Staal, 2020; Van der Ent et al., 2013). Online models track the moisture directly in an ongoing weather/-climate simulation (e.g. Arnault et al., 2021; Dahinden et al., 2023; Insua-Costa and Miguez-Macho, 2018; Koster et al., 1986; Singh et al., 2016). Alternatively, numerical models can be distinguished based on the way they formulate the transport prob-
lem (Eulerian vs Lagrangian), such as done in the review paper by Gimeno et al. (2020). Eulerian models (e.g. Arnault et al., 2021; Dahinden et al., 2023; Goessling and Reick, 2013; Insua-Costa and Miguez-Macho, 2018; Koster et al., 1986; Singh et al., 2016; Van der Ent et al., 2014) calculate concentrations and fluxes of tracked moisture in a control volume, whereas Lagrangian models (e.g., Cheng and Lu, 2023; Dey and Döös, 2020; Dirmeyer and Brubaker, 1999; Holgate et al., 2020; Keune et al., 2022; Sodemann et al., 2008; Tuinenburg and Staal, 2020) follow individual air parcels as they are transported by
the ambient fluid flow. Additionally, one could differentiate offline numerical models based on how they deal with land surface interaction: using evaporation and precipitation from the input data set (e.g., Holgate et al., 2020; Tuinenburg and Staal, 2020; Van der Ent et al., 2014), from assumptions on budget closure (e.g., Cheng and Lu, 2023; Dey and Döös, 2020; Sodemann et al., 2008), or something in between (e.g., Keune et al., 2022). Many models use simplifications, such as neglecting liquid and ice water, a reduction to 1 or 2 vertical layers, assumptions about what proportion of evaporation and rainfall is attributed to



each of the vertical layers, and the omission or parameterization of processes like microphysics, turbulent mixing and rainfall evaporation. Furthermore, one may also implement different numerical schemes and programming approaches, but we would argue that these should be considered as flavours of the same model. The effect of the differences in modelling approach and the simplifications and assumptions has received some attention in literature (e.g. Cloux et al., 2021; Crespo-Otero et al., 2024; Goessling and Reick, 2013; Li et al., 2024; Van der Ent et al., 2013) and a larger intercomparison study is underway

(Benedict et al., 2024). WAM2layers is an offline, Eulerian atmospheric moisture tracking model. Its particular simplifications, assumptions and limitations are discussed in Sections 2–4 and 6.

WAM2layers was originally developed as the single-layer Water Accounting Model (WAM v1; Van der Ent et al., 2010), driven by ERA-Interim data. The two-layer version was introduced because Van der Ent et al. (2013), who compared the capabilities of WAM to an online tracking method in a regional climate model, found that the one-layer version resulted in

large errors for cases with strong directional wind shear. After further adjustments for use with global climate and reanalysis data, the version described in Van der Ent et al. (2014, appendix B) is referred to as WAM2layers v1. Originally written in MATLAB, the first published work with a full rewrite into Python was by Van der Ent and Tuinenburg (2017, WAM2layers v2). Hence, the update to the model described here is considered to be WAM2layers v3. The corresponding code is available as van der Ent et al. (2024a). This major rewrite was aimed at improving the code efficiency in order to keep up with the increased

resolution of weather and climate model output. This latest version of the code has also seen the introduction of several best practices to increase the usability and maintainability of the software, as outlined, for example, in the practical guide to software management plans (Martinez-Ortiz et al., 2023).

Since its introduction, WAM2layers has been used by many researchers and applied to different fields. Fig. 2 shows a breakdown of all peer-reviewed papers, to our knowledge, that have actively used the model (62 papers as of September 2024).

The use of WAM2layers has increased over time and the model has predominantly been used with reanalysis data, but also sometimes with climate model output. WAM2layers has been used to gain understanding about:

– **moisture recycling** (Bedoya-Soto and Poveda, 2024; Enciso et al., 2022; Keys et al., 2014, 2024; Posada-Marín et al., 2023; Shi et al., 2022; Van der Ent et al., 2010; Zemp et al., 2014; Zhang et al., 2020),

– **climate dynamics** (Carr and Ummenhofer, 2024; Chen et al., 2023; Cluett et al., 2021; Guo et al., 2019; Li et al.,

2021, 2022; Link et al., 2020; Lobos-Roco et al., 2022; Mu et al., 2023; Van der Ent and Savenije, 2013; Van der Ent and Tuinenburg, 2017; Xia et al., 2022; Xiao and Cui, 2021; Zhang, 2020b; Zhao et al., 2016),

– **water resources governance** (Keys et al., 2017; Keys and Wang-Erlandsson, 2018; Keys et al., 2018; Posada-Marín et al., 2024),

– **vegetation dynamics** (Ampuero et al., 2020; Duerinck et al., 2016; Keys et al., 2016, 2022; Van der Ent et al., 2012, 2014),

– **land-use change** (De Hertog et al., 2024; Keys et al., 2012; Li et al., 2017; Wang-Erlandsson et al., 2018; Zemp et al., 2017),

– **paleoclimate** (Bosmans et al., 2020),



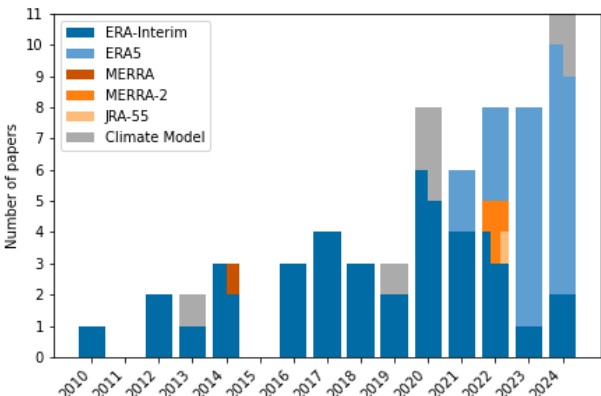

**Figure 2.** Peer-reviewed scientific papers running WAM2layers (or its predecessor WAM). Papers using WAM2layers output from other papers are not included. The year 2024 includes data until 30 September only. The data source only refers to the atmospheric wind and humidity fields as some studies replaced the reanalysis evaporation or precipitation with other observation-based gridded data sets. The category 'Climate Model' groups various different models. The underlying data is provided as supplement.

- **climate model evaluation** (Franco-Díaz et al., 2024; Guo et al., 2020),

- **climate change** (Benedict et al., 2020; Findell et al., 2019; Liu et al., 2022, 2024; Yuan et al., 2023a; Zhang et al., 2019, 2023a, b),

- **droughts** (Benedict et al., 2021; Mu et al., 2021; Pranindita et al., 2022; Zhang, 2020a; Zhou and Shi, 2024),

- **extreme precipitation** (Liu et al., 2021; Yuan et al., 2023b; Zhang et al., 2023b, 2024)

- **moisture tracking method differences** (Li et al., 2024; Van der Ent et al., 2013).

Besides, there are also studies that use the output data from WAM2layers for further analysis (Al Hasan et al., 2021; Berger et al., 2014, 2018; Cui et al., 2022; Link et al., 2021; Van der Ent and Savenije, 2011; Weng et al., 2019). Since these studies have not run the model themselves, these are not included in Fig. 2

Evidently, WAM2layers has developed into a mature piece of research software with a considerable user base. We envision that this reference manuscript will help to run it consciously with appropriate model settings, to interpret existing results, and that it will serve as a sound basis for future model improvements and sensitivity studies to understand the effects of certain modelling decisions in detail. The governing equations are derived in Section 2. Section 3 describes the pre-processing procedure, i.e., how to prepare data from various sources for a tracking experiment. Section 4 describes the actual tracking procedure. In Section 5 we briefly discuss two reference cases. Section 6 is dedicated to the best practices that have been incorporated in WAM2layers. Strengths and weaknesses, as well as opportunities for future development and best practices introduced in the development process are discussed in Section 7.



## 2 Governing equations

Here, we describe the budget equations for total and tagged moisture as they are implemented in WAM2layers v3. The formulation in this section is valid for two layers, but also extends to more layers. This facilitates comparison with other (moisture tracking/tracer) models. In Sections 3 and 4 we will describe in detail how the input data is collapsed from an arbitrary number of model or pressure levels onto a 2-layer grid, and how the equations are implemented numerically.

### 2.1 Total moisture budget

As the name "Water Accounting Model" implies, WAM2layers is primarily concerned with bookkeeping the amount of water that is exchanged between grid cells. The total water vapour $S$ per unit area $\mathcal{A}$ contained in a three-dimensional grid cell of volume $\mathcal{V} = \mathcal{A}\Delta z$ is

$$S = \frac{1}{\mathcal{A}} \int_{\mathcal{V}} q\rho \, \mathrm{d}\mathcal{V} = q\rho\Delta z = \frac{q\Delta p}{g} \tag{1}$$

where $q$ is the specific humidity, $\rho$ is the density of the air, $g$ is the gravitational constant, and $p$ is pressure. The change from height to pressure coordinates is accomplished through substitution of hydrostatic balance, which conveniently eliminates the density. Changes of grid cell area with height are neglected. Notice that WAM2layers uses specific humidity as a proxy for specific water content. As discussed in Section 3, it is possible to account for other phases of water to some extent by using the total column water, if this variable is available in the input data.

The area-averaged moisture content is convenient to work with, as the corresponding unit of $\mathrm{kg\,m^{-2}}$ is commonly used and allows for straightforward interpretation of the values of the moisture fields. Older versions of WAM2layers have worked with $(S\mathcal{A})/\rho_{lw}$ instead, where $\rho_{lw}$ represents the density of liquid water, so effectively working in units of $\mathrm{m^3}$. While the moisture balance is not affected, the input and output of WAM2layers are different between v3 and older versions.

Since the total moisture is a conserved quantity, the change of (area-averaged) moisture $S$ in a grid cell is equal to the fluxes through the boundary of the grid cell plus the sources and sinks,

$$\frac{\Delta_t S}{\Delta t} + \frac{1}{\mathcal{A}} \left( \Delta_x F_x + \Delta_y F_y + \Delta_p F_p \right) = E - P \tag{2}$$

where the terms on the left hand side represents the change of moisture in a grid cell and the horizontal and vertical fluxes respectively. The terms on the right side represent the rates of removal by precipitation, $P$, and addition through evaporation, $E$, both per unit area. Considering advection as the main mode of transport, the horizontal fluxes between grid cells are given by

$$F_x = \frac{uq\Delta y\Delta p}{g} \quad , \quad F_y = \frac{vq\Delta x\Delta p}{g} \tag{3}$$

where $u$ and $v$ are the zonal and meridional wind components, respectively. For our two-layer model, the vertical fluxes $F_p$ at the top and bottom boundaries vanish, leaving only one vertical transport term at the interface between the two layers. Since the vertical flux is positive downwards, this term acts as a sink for the top layer, and a source for the bottom layer. Importantly,





this vertical transport term is ill-defined. Advection is typically not the main process for vertical transport. Unresolved and non-isobaric processes such as convection and transport associated with precipitation can also lead to a strong vertical redistribution of water. Another complicating factor is that the vertical extent of the grid cells may vary in time and space. Therefore, in solving for (2), all terms are estimated from the input data (see Sect. 3), except for the vertical transport, which is calculated as a closure term (see Sect. 4).

## 2.2 Tagged moisture budget

The core of WAM2layers consists of tracking a certain proportion of the total moisture, which we refer to as 'tagged' moisture. We define the tagged moisture contained in a grid cell as

$$S^* = cS \tag{4}$$

where $c$ represents the tagged moisture concentration. Seeking a budget equation for $S^*$, we recast (2) in terms of tagged

moisture,

$$\frac{\Delta_t S^*}{\Delta t} + \frac{1}{\mathcal{A}} \left( \Delta_x F_x^* + \Delta_y F_y^* + \Delta_p F_p^* \right) = E^* - P^* \tag{5}$$

As opposed to (2), which is complete in terms of the input data, (5) is incomplete and must be modelled. Our quest is thus to express the unknown fluxes of tagged moisture, denoted by asterisks, as a function of the known terms of the total moisture budget. For the sources and sinks we write

$$E^* = \delta_e E \tag{6}$$

$$P^* = \delta_p P \tag{7}$$

Here, $\delta_e$ and $\delta_p$ control where water is added or removed from the system. In case of forward tracking, $\delta_e$ is 1 inside the tagging (source) region and 0 elsewhere, and $\delta_p = c$, i.e., precipitation is removed proportionally to the total moisture in the upper and lower layer. For backward tracking, it is the exact opposite: $\delta_p$ is 1 inside the tagging (sink) region, and 0 elsewhere, and $\delta_e = c$,

i.e., evaporation is removed proportionally from the lower layer.

For the fluxes, WAM2layers uses

$$F_x^* = cF_x \qquad F_y^* = cF_y \qquad F_p^* = cF_p + \kappa \frac{\Delta_p c}{\Delta p} \tag{8}$$

that is, tagged moisture transport is proportional to the transport of total moisture in the horizontal direction. In the vertical, an additional dispersion term is added, since Van der Ent et al. (2014) noticed that there was insufficient vertical mixing of tagged

moisture without it. Setting $\kappa = k_{vf}|F_p|\Delta p$ ensures that the dispersion rate is proportional to vertical advection rate $F_p$ and the grid size. A similar form may be found in Cushman-Roisin and Beckers (2011, p. 138), who reason that $\kappa$ is characterized by the largest unresolved eddies. Here, $k_{vf}$ is a tunable parameter to control the dispersion rate. Van der Ent et al. (2014) suggested that a value of 3 gives realistic results, but this could be case-dependent.



## 2.3 Some notes on the formulation of WAM2layers

Though originally conceived as a simple bookkeeping exercise, the accounting method in WAM2layers has a formal basis in the finite volume approach. In the absence of sources and sinks, conservation of moisture reads

$$\frac{\partial(\rho q)}{\partial t} + \nabla \cdot \boldsymbol{f}(\rho q) = 0 \tag{9}$$

where $\boldsymbol{f}(x)$ represents a generic flux function (for advection only, $\boldsymbol{f}(x) = \boldsymbol{u}x$). Integrating all terms over a fixed control volume and applying Gauss' theorem to the divergence term yields an equivalent expression, stating that the rate of change of moisture

in a grid cell is equivalent to the flux across its enclosing surface. Upon discretization, substitution of hydrostatic balance, and division by the grid cell area, we retrieve (2). Alternatively, starting isobaric coordinates to yields the same result. Applying this procedure to grid cells with a variable vertical extent rather than a fixed control volume yields an additional term by virtue of Leibniz' rule. This term is effectively 'absorbed' by our loose definition of the vertical flux.

On a Cartesian grid with $\mathcal{A} = \Delta x \Delta y$, the horizontal transport in terms in (2) can be written as $\nabla \cdot \boldsymbol{F}$ with

165 $$\boldsymbol{F} = (F_x, F_y) = \left( \frac{uq\Delta p}{g}, \frac{vq\Delta p}{g} \right) \tag{10}$$

This testifies to the equivalence of the continous and integrated budget equations. However, since WAM2layers operates on a spherical latitude/longitude grid, we must stick with (2) to ensure meridional transport is dependent on latitude.

To facilitate numerical stability analysis and comparison to other moisture tracking methods and, more broadly, to tracer models in general, it is convenient to define an 'integrated moisture velocity' $\hat{\boldsymbol{u}} = (\hat{u}, \hat{v}, \hat{\omega})$ such that

$$F_x = \hat{u}S \qquad F_y = \hat{v}S \qquad F_p = \hat{\omega}S \tag{11}$$

With this, conservation of total moisture, tagged moisture, and tagged moisture concentration can all be expressed in the same form:

$$\frac{\partial S}{\partial t} + \nabla \cdot (\hat{\boldsymbol{u}}S) = 0 \tag{12}$$

$$\frac{\partial S^*}{\partial t} + \nabla \cdot (\hat{\boldsymbol{u}}S^*) = 0 \tag{13}$$

$$\frac{\partial c}{\partial t} + \nabla \cdot (\hat{\boldsymbol{u}}c) = 0 \tag{14}$$

In relation to existing literature, we point out that Trenberth and Guillemot (1995); Dominguez et al. (2006); Burde and Zangvil (2001a) take a slightly different approach to calculate integrated vapour transport. They take the vertical integral of each term in (9) in isobaric coordinates and apply Leibniz' rule to each of the terms, upon which the vertical flux disappears from the budget equation. In their case, since they consider the budget equation vertically integrated over the full atmosphere,

this makes perfect sense. In our case, as mentioned before, we need to retain a vertical transport term between the two layers of our model, due to unresolved and non-isobaric processes, and variability of the grid cells' vertical extent. Note that the aforementioned studies used the symbol $w$ (sometimes referred to as 'precipitable water') for what we refer to as the total moisture $S$.



## 2.4 Summary

Recapitulating, the heart of WAM2layers consists of the equations (2) and (5) for total moisture $S$ and tagged moisture $S^*$, respectively. These governing equations naturally lead to a two-step process for using WAM2layers. First, the input data are collapsed onto a two-layer grid, and the terms in (2) are reconstructed from the available input data . This is what we refer to as the pre-processing step. Subsequently, (5) is integrated numerically, which we call the tracking step. In the next sections, we will discuss each of these steps in detail.

## 3 Pre-processing

The first step in any WAM2layers experiment is to collapse the input data onto a two-layer grid and to reconstruct the corresponding terms in (2) for each new grid cell, except for the vertical flux, which will be calculated as a closure term later on. This procedure is dataset-dependent, and therefore it is isolated from the rest of the code in a dedicated pre-processing module. This makes it possible to support multiple input datasets with minimal code duplication.

In developing WAM2layers v3, we have primarily focused on working with ERA5 reanalysis data (Hersbach et al., 2020). The comprehensiveness of this dataset enables scrutiny in the treatment of vertical levels, which also facilitates the presentation in this manuscript. We aim for a complete description such that a similar procedure can easily be derived for other data sources.

### 3.1 Data retrieval: required variables

ERA5 data can be downloaded at model levels or pressure levels. In either case, the forthcoming steps require as input the horizontal wind components $u$ and $v$, as well as specific humidity $q$, at all or a subset of the available levels. Furthermore, precipitation ($P$) and evaporation ($E$), as well as surface pressure ($p_s$) are required at the surface, and we also include total column water ($S_{tc}$). This last variable is not strictly required to run WAM2layers, but it enables us to estimate and mitigate (to some extent) the errors introduced by the negligence of liquid and ice water. Working with pressure level data additionally requires the surface values of wind and water, i.e., 10-meter wind components $u_{10}$ and $v_{10}$ and 2-meter dewpoint temperature $T_{d,2}$.

### 3.2 States and fluxes

We start by describing the ideal case where the three-dimensional fields of $u$, $v$, and $q$ are available on all model levels. WAM2layers aggregates the input data onto a two-layer grid by partitioning the input grid in an upper ($k = 0$) and lower ($k = 1$) part, and taking the sum over each layer.

Summation of (1) over each partition $k$ gives the total water in each grid cell in WAM2layers,

$$S_k = \frac{1}{g}\sum_{i\in k} q_i \Delta p_i \tag{15}$$





Here the subscript $i$ refers to indices in the ERA5 level definition, which can be found in the documentation of the underlying Integrated Forecasting System (IFS) (ECMWF, 2016). Horizontal advection terms are calculated according to (3), and aggregated in a similar fashion

$$F_{x,k} = \frac{1}{g}\sum_{i \in k} u_i q_i \Delta p_i \quad , \quad F_{y,k} = \frac{1}{g}\sum_{i \in k} v_i q_i \Delta p_i \tag{16}$$

Following Van der Ent et al. (2013, 2014), we set the boundary to be at the interface between IFS levels 111 and 112 by default. This corresponds to around $812\,\mathrm{hPa}$, for a standard atmospheric pressure of $1013.25\,\mathrm{hPa}$. This boundary in WAM2layers is usually just above the atmospheric boundary layer with on average slightly more moisture below the boundary than above the boundary. Note that the boundary can be at a much lower pressure over mountainous terrain.

The vertical pressure difference over each input grid cell can be calculated with

$$\Delta p_i = p_{i+1/2} - p_{i-1/2} \tag{17}$$

$$p_{i+1/2} = A_{i+1/2} + p_{s,i+1/2}B \tag{18}$$

where $A$ and $B$ are hybrid model level coefficients describing the vertical discretization of the model levels in IFS (ECMWF, 2016).

**Accounting for liquid and solid water**

Equation (15) represents the total mass of water *vapour* in each grid cell. By using the total column water, which is also included in the ERA5 output variables, we can apply a rather ad-hoc adjustment to account for other phases of water:

$$S_k = S_k \frac{Q_{tc}}{\sum_k S_k} \tag{19}$$

This effectively distributes cloud and rain/snow water across all levels, which is not perfect, but better than not counting it at all. For the two example cases, this correction is typically within 1 % when averaged over the grid, but may be more than 10 % for specific individual grid cells and time steps. WAM2layers provides information about this in the onscreen logging and associated logfile.

### 3.3   Sources and sinks

In the formulation of WAM2layers's equations we adopted the common convention in hydrology of having only positive values for both precipitation and evaporation. Precipitation represents all water transfer from the atmosphere to the surface, including condensation.

In ERA5, vertical fluxes are defined positive downwards. The evaporation variable can be either positive (condensation) or negative (evaporation). Therefore, during the pre-processing step for ERA5, the sign of evaporation is flipped and any remaining negative values are re-assigned to the precipitation variable instead.

In a forward-tracking experiment, a proportion of the total surface evaporation is tagged ($\delta_e$). Similarly, for a backward-tracking experiment, we tag a proportion of the precipitation ($\delta_p$). In either case, an additional input file is prepared that defines





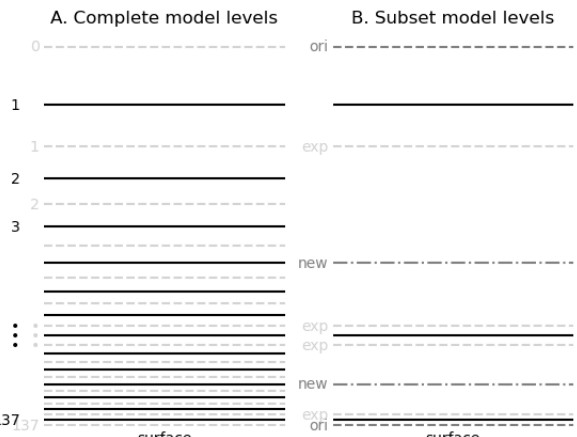

**Figure 3.** Illustration of how WAM2layers treats a subset of layers. Full black lines are full-levels, dashed grey lines are half-levels. In B, the top and bottom half-levels are retained, but the remaining half-levels are expired. New half-levels are obtained by interpolating between the selected full-levels.

the tagging region. This file contains a two- or three-dimensional mask of values between 0 and 1. The three-dimensional mask, which is still relatively experimental, would allow for a moving tagging region, e.g. for studying storms. In the two-dimensional case, a tagging start- and end date are given in the configuration file.

### 3.4 Dealing with fewer model levels

ERA5 data is available at all 137 model levels, which is quite unique in terms of resolution (Hersbach et al., 2020), but also quite demanding in terms of storage and processing. Since we collapse the data onto a two-layer grid anyway, it is not necessary to use all 137 levels. To use WAM2layers with only a subset of these 137 layers, the $A$ and $B$ in (18) are interpolated first to the full-levels of the layer subset, then to the interfaces between these full-levels. This procedure is illustrated in Figure 3.

Based on our experience with a few test cases, around 20 layers gives good results, provided that the sampling is dense enough near the surface where most of the moisture is located. The number of input levels is configurable though, so users of WAM2layers are encouraged to experiment with different subsets as they see fit.

### 3.5 Working with pressure-level data

Working with (a limited number of) pressure levels instead of model levels is sometimes the only option. This is, for example, the case for datasets from CMIP archives (Juckes et al., 2020, Table 4). ERA5 data is also available on pressure levels, and WAM2layers v3 provides code to work with pressure level data as well.

From a point of view of achieving accurate moisture tracking having wind and humidity data near the surface is crucial, but this could be absent in case of limited pressure levels. Therefore, in order to work with (ERA5) pressure-level data, we complement the data with additional values from the surface-level data (2-meter specific humidity and 10-meter wind) as these



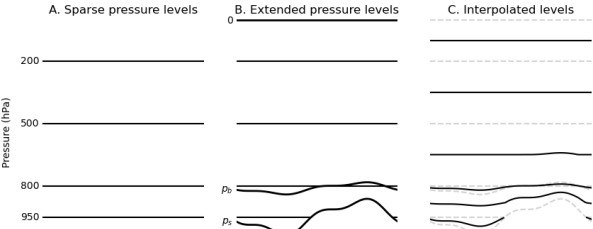

**Figure 4.** Pressure-level data pre-processing. A: original pressure levels. B: The original levels are extended at the top, surface, and the interface is inserted. C: Wind and humidity are interpolated to the midpoints between the pressure coordinate, to become the new full levels (full black lines). Pressure is kept at the original levels, which thus become the new half-levels (dashed grey lines).

are generally more representative of the surface conditions than the closest available pressure level. In addition, at the top of the atmosphere, WAM2layers assumes humidity to be zero and wind equal to the pressure level with the lowest pressure. This procedure is illustrated in Figure 4.

The surface pressure is used to insert these values in between the right pressure levels. Since surface pressure is space and time dependent ($p_s = p_s(x, y, t)$), the same now applies to our vertical coordinate: $p = p(x, y, z, t)$. A further complication is that the pressure levels can intersect the surface, so we mask any data where the pressure is higher than the surface pressure.

The interface between the two layers in WAM2layers must also follow the terrain. For consistency with the treatment of model-level data, we define the interface according to (18). By default, we use the $A$ and $B$ values for level IFS 111, but since WAM2layers v3.1 this is configurable so users of WAM2layers can easily modify it as they see fit.

At this point, the variables of interest, wind speed and humidity, are collocated with the pressure coordinate. To calculate the grid cell moisture content and fluxes, we need to multiply the pressure difference over a volume with representative values for that same volume. Thus, we interpolate wind and humidity to the interfaces between the pressure coordinates. Finally, we can proceed with evaluating (15) and (16) as before.

### 3.6 Pre-processing other datasets

WAM2layers is set up in such a way that it is relatively easy to extend its pre-processsing to any dataset. Most of the steps outlined above are implemented as generic routines that can be used for any dataset. To add a dataset, only a data loader script has to be implemented, which loads data into a standard internal format. As of version 3.1, built-in pre-processing is available for ERA5 on both model levels and pressure levels, and preprocessing for CMIP data (on pressure levels) is being developed for v3.2. We envision that more datasets will be included in the future, and we foster community contributions such that other researchers can also contribute their own custom pre-processing scripts.





## 4 Tracking

The pre-processing step is designed in such a way that the intermediate data can be stored efficiently. As part of the tracking routine, the data is first interpolated to a finer time step, after which we solve for the vertical transport term in (2). Finally, (5) is integrated numerically.

### 4.1 Solving for the vertical flux

In discretizing (2), we assign the full evaporation source to the lower layer, but we distribute the precipitation sink proportionally over both layers. Additionally, we introduce an error term to account for inaccuracies, numerical errors, and inconsistencies in the input data caused by both the raw (ERA5) data as well as the pre-processing in WAM2layers. This leads to the following, fully discretized system of equations

$$
\begin{aligned}
\frac{S_{ijk}^{t+1} - S_{ijk}^t}{\Delta t} &+ (F_{x,i+1/2jk} - F_{x,i-1/2jk})/\mathcal{A}_j \\
&+ (F_{y,ij+1/2k} - F_{y,ij-1/2k})/\mathcal{A}_j \\
&+ (F_{p,ijk+1/2} - F_{p,ijk-1/2})/\mathcal{A}_j \\
&= \delta_k E_{ij} - \frac{S_{ijk}}{S_{ijt}} P_{ij} + \epsilon_{ijk}^{t+1}
\end{aligned}
\tag{20}
$$

where all fluxes are evaluated at $t + 1/2$; the superscripts are omitted for clarity, $S_{ijt}$ represents the total column water, and $\delta_k$ is 1 for the lower layer ($k = 1$) and 0 otherwise. Note that in (20) the vertical flux is positive downward in accordance with the pressure coordinate. For convenience we can move all known terms to the left-hand side and all unknowns to the right. Using $R$ to denote the known terms and leaving out the subscripts $i, j$ for clarity, we get

$$
R_k = -(F_{p,k+1/2} - F_{p,k-1/2})/\mathcal{A} + \epsilon_k
\tag{21}
$$

After applying the boundary conditions $F_p = 0$ at the bottom and top, the only remaining net vertical flux in our two-layer model is at the interface $F_p = F_{p,1/2}$, such that

$$
R_0 = -\frac{F_{p,1/2}}{\mathcal{A}} + \epsilon_0 \quad , \quad R_1 = +\frac{F_{p,1/2}}{\mathcal{A}} + \epsilon_1
\tag{22}
$$

To solve this system, we introduce another constraint by requiring that the errors are proportionally distributed between the upper and lower layer:

$$
\frac{\epsilon_k}{\epsilon_t} = \frac{S_k}{S_t}
\tag{23}
$$

where subscript $t$ denotes the column totals. Isolating $\epsilon_k$ and recognizing that the total error should be equal to the total residual, we have

$$
\epsilon_k = \frac{\epsilon_t S_k}{S_t} = \frac{R_t S_k}{S_t}
\tag{24}
$$



Substituting this result in (22) we finally arrive at

$$\frac{F_{p,1/2}}{\mathcal{A}} = -R_0 + \frac{R_t S_0}{S_t} = R_1 - \frac{R_t S_1}{S_t} \tag{25}$$

In the software implementation, as well as in previous publication, this term $F_{p,1/2}/\mathcal{A}$ is typically simply referred to as 'the vertical flux', $F_v$.

## 4.2 Numerical integration

The formulation of WAM2layers naturally lends itself to a finite-volume implementation, where the incoming fluxes in one cell correspond to outgoing fluxes in its neighbours. Concretely, in case of forward tracking, we have for the upper and lower layers, respectively,

$$
\begin{aligned}
S_{ij0}^{t+1} = S_{ij0}^t + \Delta t \big[ \\
&- (F_{x,i+1/2,j0}^* - F_{x,i-1/2,j0}^*)/\mathcal{A} \\
&- (F_{y,i,j+1/2,0}^* - F_{y,i,j-1/2,0}^*)/\mathcal{A} \\
&- (F_{p,ij,1/2}^* + k_{vf}|F_{p,ij,1/2}|(c_{ij,1}^t - c_{ij0}^t))/\mathcal{A} \\
&- c_{ij0}\frac{S_{ij0}}{S_{ijt}}P_{ij}^{t+1/2}\big]
\end{aligned}
\tag{26}
$$

$$
\begin{aligned}
S_{ij1}^{t+1} = S_{ij1}^t + \Delta t \big[ \\
&- (F_{x,i+1/2,j1}^* - F_{x,i-1/2,j1}^*)/\mathcal{A} \\
&- (F_{y,i,j+1/2,1}^* - F_{y,i,j-1/2,1}^*)/\mathcal{A} \\
&+ (F_{p,ij,1/2}^* + k_{vf}|F_p|(c_{ij,1}^t - c_{ij0}^t))/\mathcal{A} \\
&+ \delta_e E_{ij}^{t+1/2} - c_{ij1}\frac{S_{ij1}}{S_{ijt}}P_{ij}^{t+1/2}\big]
\end{aligned}
\tag{27}
$$

Notice that the vertical transport terms are directed from the upper to the lower layer, i.e., positive downward. $\delta_e$ is the tagging region (see Section3).

A crucial aspect of any finite volume scheme is how the fluxes at the interfaces are calculated. WAM2layers employs a simple donor cell scheme (see, e.g. Cushman-Roisin and Beckers, 2011). In this scheme, the flux of tagged moisture $F^* = cF$ is constructed by combining $F$, calculated at the interfaces of the grid cell and at $t + 1/2$, with $c$ at $t$, taken from the *upstream* volume. Concretely,

$$
(cF)_{x,i+1/2jk} = \begin{cases} F_{x,i+1/2jk}^{t+1/2}\, c_{ijk}^t, & \text{if } F_{x,i+1/2jk}^{t+1/2} > 0 \\ F_{x,i+1/2jk}^{t+1/2}\, c_{i+1jk}^t, & \text{otherwise} \end{cases}
\tag{28}
$$

and similarly for the other directions.





In case of backward tracking, the equations are similar,

$$
\begin{aligned}
S_{ij0}^{t-1} = S_{ij0}^t - \Delta t \big[ \\
- (F_{x,i+1/2,j0}^* - F_{x,i-1/2,j0}^*)/\mathcal{A} \\
- (F_{y,i,j+1/2,0}^* - F_{y,i,j-1/2,0}^*)/\mathcal{A} \\
- (F_{p,ij,1/2}^* + k_{vf}|F_p|(c_{ij,1}^t - c_{ij0}^t))/\mathcal{A} \\
- \delta_p \frac{S_{ij0}}{S_{ijt}} P_{ij}^{t-1/2} \big]
\end{aligned}
\tag{29}
$$

$$
\begin{aligned}
S_{ij1}^{t-1} = S_{ij1}^t - \Delta t \big[ \\
- (F_{x,i+1/2,j1}^* - F_{x,i-1/2,j1}^*)/\mathcal{A} \\
- (F_{y,i,j+1/2,1}^* - F_{y,i,j-1/2,1}^*)/\mathcal{A} \\
+ (F_{p,ij,1/2}^* + k_{vf}|F_p|(c_{ij,1}^t - c_{ij0}^t))/\mathcal{A} \\
+ c_{ij1}E_{ij}^{t-1/2} - \delta_p \frac{S_{ij1}}{S_{ijt}} P_{ij}^{t-1/2} \big]
\end{aligned}
\tag{30}
$$

Effectively, all fluxes change direction, which means that the condition in (28) should also be reversed. Also note that the tagged moisture mask is now applied to precipitation instead of evaporation.

## 4.3 Stability considerations

Here we briefly consider the timestep at which WAM2layers should operate. For an explicit solver, a common criterion is that the Courant number should not exceed unity. In other words, water should not be allowed to move more than one grid cell per time step. To explore this limit, we can use the moisture velocity defined in (11).

Figure 5 shows the corresponding maximum time step for a typical range of moisture velocity and the ERA5 grid. It shows that, due to the convergence of the meridians, the maximum time step rapidly decreases towards the poles. For this reason, we strongly advise against using WAM2layers on input data that extends beyond $\sim 75°$ N/S. Even for tracking experiments with a tagging region located far away from the poles, it is wise to exclude the poles from the calculation domain to avoid spurious results. For other input datasets, a similar assessment should always be made.

In principle, the time step should be chosen sufficiently small so as to satisfy the CFL criterion. However, choosing a very small time step increases computation time and exacerbates numerical diffusion. So there is an incentive to choose a time step that is sufficiently small for, say, $\sim 95\,\%$ of the time, and accept that there may be some extreme circumstances where the CFL criterion would be violated. In this case, however, we do need to constrain the fluxes for these extremes. To this end, WAM2layers imposes some limits such that the combined moisture fluxes will at most be able to empty the upper or lower grid cell completely. The first limit is such that the meridional and zonal fluxes combined can at most empty the grid cell, but not create any numerical water gains. The second limit is such that a such that the gross vertical flux can also at most empty the grid cell. These limits are not $100\,\%$ watertight in case both the horizontal and vertical fluxes are of great magnitude, which





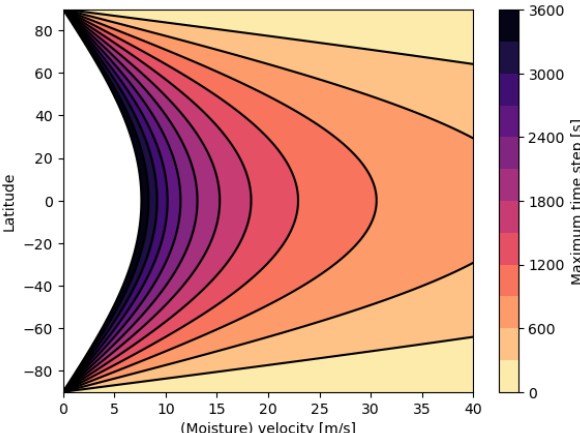

**Figure 5.** Timestep for which the Courant number resolves to unity, based on grid spacing of 0.25°as in the ERA5 input data and a typical range of moisture velocities seen in the example cases.

can theoretically result in numerical tagged moisture gains. However, this is being monitored in the logging and should warrant a reconsideration of the time step.

At the end of each time step, WAM2layers checks against moisture surplus or deficit in any grid cell. Notably, tagged moisture cannot exceed the total moisture in a grid cell. Should there be any imbalance, WAM2layers attempts to redistribute
the moisture between the upper and lower layer. If this is not sufficient, tagged moisture is lost from the system internally. In v3, we have added log messages to monitor the ongoing experiment, and the spatial fields are included in the output for later analysis. This provides confidence as long as everything is okay, and highlights problems early if they arise. Similary, WAM2layers logs and report losses over the boundary of the domain, which are naturally expected, but in case this boundary transport is significant one might consider increasing the size of the domain.

**5   Example use cases**

Here, we present two example cases to show some of the possibilities of WAM2layers v3. We selected one forward and one backward tracking example with different event duration. The input data as well as the configuration for these cases are available via the 4TU.ResearchData data repository (Benedict and Weijenborg, 2024; Gaasbeek and van der Ent, 2024), and WAM2layers implements a download utility to automatically retrieve these datasets by their DOI. As such, anyone can easily
reproduce these example cases and get started with WAM2layers.

The backward tracking example case determines the moisture sources of the extreme precipitation event over the Eiffel region in western Europe (Belgium, France, Germany, Luxemburg and the Netherlands) on 13-14 July 2021. This was a catastrophic event where extreme precipitation over two days resulted in large floods in the tributaries of the Lower Rhine, such as the Ahr, Erft and Wupper and the river Meuse, such as the, Ourthe, Rur and Geul (Kreienkamp et al., 2021; Mohr





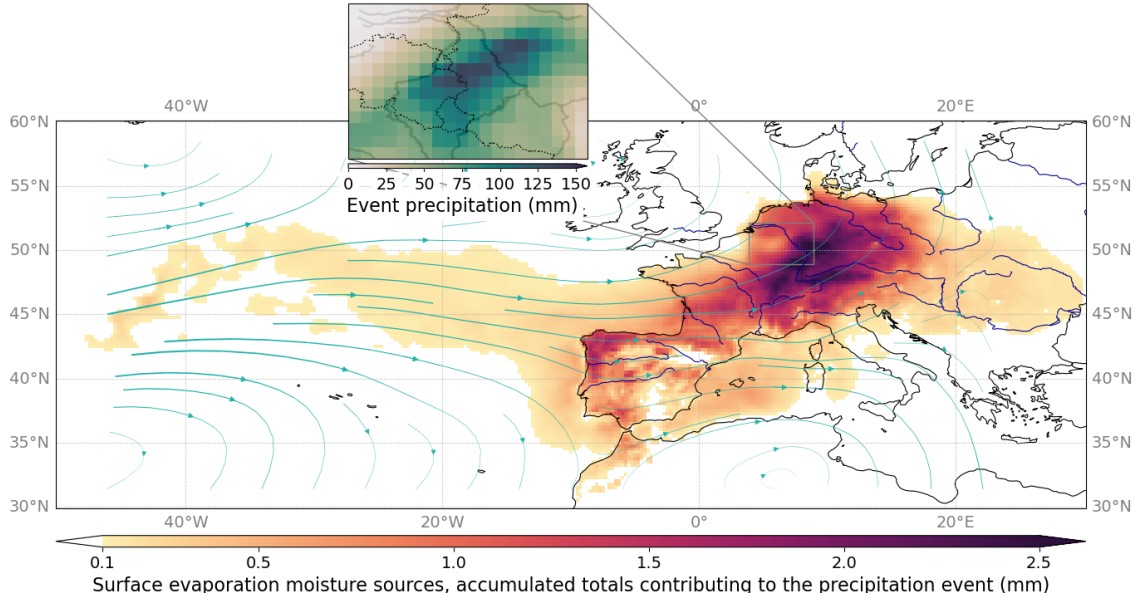

**Figure 6.** Outcome of the backward tracking example case 'Eiffel'. The figure shows the accumulated tracked moisture sources of the extreme precipitation event of 13-14 July 2021. Total precipitation over the sink region during the event is displayed as inset. Backward tracking was performed until 1 July. By then, 42 % of moisture was tracked to its source; 3.4 % of was still in the domain's atmosphere, and 54 % was associated with transport across the domain boundaries. Streamlines depict the average vertically integrated moisture flow during 1-14 July.

et al., 2023). The moisture sources of this event have been previously quantified using different moisture tracking models as shown in Insua-Costa et al. (2022) and Staal and Koren (2023) and now by WAM2layers in Fig. 6. The results show that the evaporative sources of this heavy precipitation event were mostly located over Germany and France. Besides, there is a substantial contribution from the Atlantic ocean and a small contribution from the Mediterranean Sea. The spatial pattern of the sources corresponds to the sources determined by Insua-Costa et al. (2022) and Staal and Koren (2023).

The forward tracking example case tracks evaporation in the Lake Volta region (Ghana) forward in time for July 1998 to identify the precipitating sinks (Fig. 7). This case was also used in a comparison of moisture tracking models (Van der Ent et al., 2013), although it is not exactly the same. Input data for the previous study was from the regional climate model MM5 (Knoche and Kunstmann, 2013), the domain was smaller and the evaporation was tagged during August as well. Nonetheless, we can observe the similar moisture source patterns. These patterns are caused by the southwesterly winds transporting moisture in the

lower levels, until the African Easterly Jet picks up the tagged moisture and transport it back in westerly direction towards the Atlantic. Note that the average vertically integrated moisture fluxes alone as shown by the streamlines in Fig. 7 cannot capture the complicated moisture transport in this system with strong wind shear and as such highlight the importance of having two layers in WAM2layers.



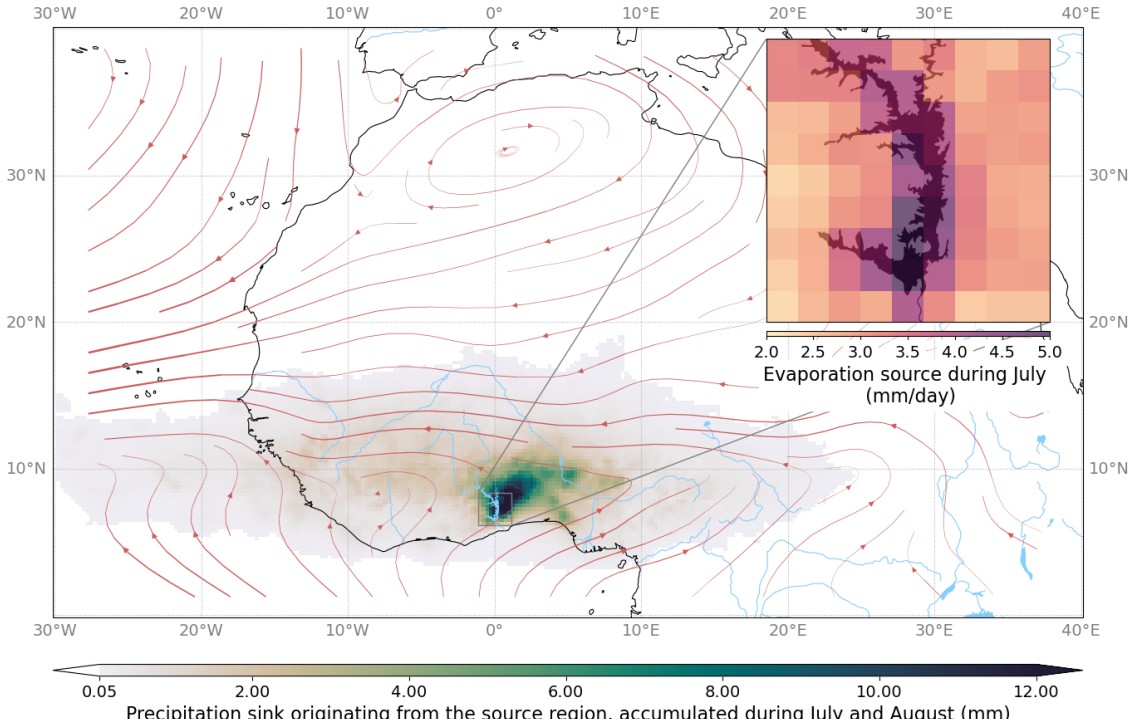

**Figure 7.** Outcome of the forward tracking example case 'Volta'. The figure shows the tracked moisture sinks of the evaporation from the lake Volta region during July 1998 while the tracking continued until the end of August 1998. The inset shows the time-averaged evaporation in the source region. 86 % of the total tagged evaporated moisture was attributed to surface precipitation and 14 % of the tagged moisture was transported across the boundaries of the domain. Of the tagged moisture 7.8 % recycled within the source region. Streamlines depict the average vertically integrated moisture flow during July and August.

## 6   Best practices in research software development

During the development of WAM2layers v3, we have put a lot of emphasis on the incorporation of research software engineering best practices aimed at making WAM2layers more user-friendly, easier to maintain, and to enable a more open and transparent scientific process (Barker et al., 2022; Martinez-Ortiz et al., 2023).

One of the key innovations is that WAM2layers has been turned into an installable command-line program. Users no longer need to edit the source code in order to be able to configure their model. Instead, they simply edit a configuration file. This

file contains all settings of the model, including the direction of tracking (forward or backward), the internal time step of the model, the interval on which the output data is provided, and the dates to perform the tracking for. This makes it easier to share and reproduce experiments. To further enhance reproducibility, WAM2layers creates a dedicated output folder for each experiment, which includes a copy of the configuration file and a log file. Each release of WAM2layers has a corresponding



DOI, so users can always refer to the exact version of WAM2layers used for their experiments, here version 3.1 (van der Ent
et al., 2024b).

Importantly, we have added extensive user and developer documentation, which is available at https://wam2layers.readthedocs.
io. The user documentation includes a quickstart guide, which lets the user run both example cases described in section 5 in a
matter of minutes. To this end, we have made WAM2layers available as a Python package on PyPI, such that it can be installed
with a single command. Additionally, we have made sample data available at the 4TU.ResearchData repository (Benedict and
Weijenborg, 2024; Gaasbeek and van der Ent, 2024), and added functionality to automatically download these datasets. This
could easily be extended to include more reference cases, so we encourage users of WAM2layers to make their cases available
in a similar manner. To wrap up the quickstart, WAM2layers ships with a quick analysis module to quickly inspect the input
or output of an experiment, but we recommend users to use more elaborate case-specific code to produce publication-quality
figures.

Apart from technical contributions, we have invested in building a welcoming community on GitHub (https://github.com/
WAM2layers/WAM2layers/). All development happens openly and we encourage users to engage and contribute through
opening issues, pull requests, or through the discussion forum where they can ask questions, showcase their results and share
codes. The documentation includes community guidelines and instructions for developing the model, as well as an explanation
of the collaborative development process. With respect to developer experience, we formulated guidelines for code formatting,
and we have set up automated tests to ensure the continuity of the model code. It is verified that results for simple test cases do
not change upon making changes to the model code, unless this is expected based on the changes. This strategy is known as
'regression testing' (e.g., Pezze, 2008), and it can provide a lot of confidence for (new) developers.

## 7   Discussion

The Water Accounting Model has since its inception (Van der Ent et al., 2010) evolved from a loose collection of scripts to
calculate moisture recycling characteristics into a widely used atmospheric moisture tracking software with a broad range of
applications (Fig. 2). This manuscript documents WAM2layers v3 (van der Ent et al., 2024a), which forms a solid basis for
future experiments and further development. Here we discuss some key strengths, as well as aspects where we think the model
could still be improved.

### 7.1   Important simplifications

In WAM2layers, the evolution of (tagged) moisture is completely governed by precipitation, evaporation, and redistribution
through advection, vertical mixing and, artificially, numerical diffusion. Horizontal mixing, microphysics and other subgrid
processes are omitted. These simplifications, partly born of necessity, introduce inaccuracies that users of WAM2layers should
beware of. Below we discuss these simplifications in more detail and how future work may test the importance of these
simplification and potentially improve on them.





Whilst we omit horizontal (turbulent) diffusion from the governing equations (2) and (5), the donor cell scheme employed in WAM2layers (aka 'upstream' or 'Godunov' scheme) is known to be numerically diffusive itself (Hourdin and Armengaud, 1999). This leads to a situation where artificial numerical diffusion may actually exceed the true physical diffusion, which is undesirable (Cushman-Roisin and Beckers, 2011). It limits WAM2layers' usability at high latitudes and may lead to an overestimation of far away sources. Ideally, one would want to suppress or eliminate numerical diffusion altogether, or at

least get a better grip on it. One option could be to explore the use of alternative solvers, such as flux-limiters (Hourdin and Armengaud, 1999; Cushman-Roisin and Beckers, 2011) or MPDATA (Smolarkiewicz and Margolin, 1998). Once the numerical diffusion is under control, we may wish to consider the introduction of a true physical diffusion term instead. This becomes more relevant as the resolution of the input data continues to increase. However, despite its flaws, we do emphasize that the donor cell scheme has some favourable properties as well; it is simple, intuitive, conservative, and numerically stable. So in

reconsidering all this, we should be careful not to throw the baby out with the bathwater.

WAM2layers uses water vapour as a proxy for total water, and does not represent microphysical processes associated with phase changes or transport of ice and liquid water. This is a common limitation in offline moisture tracking models (e.g. Cheng and Lu, 2023; Dey and Döös, 2020; Dirmeyer and Brubaker, 1999; Holgate et al., 2020; Keune et al., 2022; Sodemann et al., 2008; Tuinenburg and Staal, 2020). As discussed in Section 3, we can sometimes partly correct the states and horizontal fluxes

for the presence of ice and liquid water by application of (19). However, condensation and re-evaporation may be most relevant in the vicinity of precipitation, in which case they could bring about a strong vertical exchange of (tagged) moisture, e.g., condensating in the upper layer and re-evaporating in the lower layer. This is one of the reasons why it is important to realize that the vertical transport term, which is calculated as a closure term in WAM2layers, effectively acts as a generic redistribution term, incorporating the effects of unresolved processes such as microphysics and (turbulent) diffusion, and even compensating

for discontinuities in the input data or errors in the horizontal fluxes.

Finally, a note on the reduction to two layers. Van der Ent et al. (2013) found that a two-layer model was both necessary and sufficient to represent large-scale atmospheric moisture transport. The two-layer model simplifies the computation of the vertical moisture flux and the partitioning of precipitation between the layers of the model. However, it also leads to a two-step procedure where all three-dimensional fields first need to be aggregated onto the new grid. Generalizing to $N$ layers could

potentially simplify the code. However, it remains to be seen whether this would actually improve the results, as a correct representation of all vertical exchange processes would remain challenging and should be thoroughly tested.

## 7.2   A note on performance

WAM2layers v3 is written in Python. This programming language is very popular among (atmospheric) scientists, which makes it easy to use. Moreover, Python's simple syntax makes it easy to understand the models internal workings, which we deem

key from a transparency point-of-view. However, Python has somewhat of a poor reputation when it comes to computational performance. Evidently, the use of numerical computing libraries such as NumPy alleviates this to a certain extent (Harris et al., 2020; Langtangen and Cai, 2008). Further performance gains could potentially be achieved by using dedicated libraries for stencil computations, or by porting some of the performance critical code to more performant languages. A potential downside



of using experimental third-party libraries or mixing languages is that it puts a bigger constraint on the maintainability of the
software. Treading this path therefore remains a balancing act, and we tend to favour usability and code transparency over
performance.

### 7.3 Future developments

Software development is planned to continue also after the WAM2layers release associated with this manuscript (version 3.1.0
at the time of writing). Future developments notably aim to bring back some old features that have not yet been ported to
version 3, such as quick regional moisture recycling calculations (e.g., Van der Ent and Savenije, 2011; De Hertog et al.,
2024), time tracking (e.g., Van der Ent et al., 2014; Van der Ent and Tuinenburg, 2017), distance tracking (e.g. Guo et al.,
2019, 2020) and advanced spline interpolation of the humidity profile in case of limited vertical information (Benedict et al.,
2020). We are planning to re-incorporate these features in a backwards-compatible manner. Similarly, we would like to port
pre-processing codes for other reanalyses datasets (e.g., Keys et al., 2024; Li et al., 2022) as well as climate model output with
which WAM2layers has been used in the past (e.g., Bosmans et al., 2020; De Hertog et al., 2024; Findell et al., 2019; Guo
et al., 2020) to future model releases as well.

In developing WAM2layers v3, we have made substantial efforts to generalize the derivation presented here, and to isolate
the implementation of the numerical scheme in the code. We believe that this manuscript, together with the changes made for
WAM2layers v3, form a good basis for modelling exercises such as implementing alternative solvers or further increasing the
performance.

Simultaneously with writing this manuscript, we are co-organizing a coordinated moisture tracking intercomparison study (Bene-
dict et al., 2024), to gain understanding of the uncertainties in different tracking methods. We expect that these community
activities will help to shed a better light on the strengths and weaknesses of WAM2layers and other models, and to gain
confidence in moisture tracking results in general.

### 8 Conclusions

To conclude, this manuscript fills an important gap in the documentation of a widely used atmospheric moisture tracking model.
We provided an updated description of the model that facilitates comparison with other models and literature, and pointed out
some important assumptions, which facilitates interpretation of past and future results obtained with WAM2layers.

This manuscript, and the associated v3 release, mark a new milestone in the evolution of WAM2layers. While the core
tracking principle and assumptions remain unchanged, the code is now much more efficient and modular, making it ready for
future developments. We have pointed out several directions for future work, including a more thorough investigation of the
numerical aspects and opportunities for improved model validation in collaboration with the moisture tracking community.

With the incorporation of various best practices as outlined in the practical guide to software management plans (Martinez-
Ortiz et al., 2023), and the engagement with the wider community, we have made important steps to carry out FAIR and open,
collaborative research software development, and we are committed to improving this further into the future. We hope this



approach will inspire others to follow suit, as we believe that transparency and collaboration are crucial for the credibility of prosperity of the scientific field.

*Code availability.* WAM2layers is available on PyPI at https://pypi.org/project/wam2layers/. The source code is distributed under a permissive Apache 2.0 license and is available on GitHub at https://github.com/WAM2layers/WAM2layers. Documentation lives at https://wam2layers.readthedocs.io/,
which includes a description to reproduce the example cases. WAM2layers can be cited via the following DOI: https://doi.org/10.5281/zenodo.7010594
.

*Data availability.* The data for the example cases is hosted at the 4TU.ResearchData repository, see Benedict and Weijenborg (2024); Gaasbeek and van der Ent (2024). The data on WAM2layers literature underlying Fig. 2 is provided as supplement to this manuscript.

*Author contributions.* PK lead the development of WAM2layers v3 and writing of the manuscript. IB, CW and RvdE provided input and
feedback on the code and the manuscript. IB lead the project under which most of the development of WAM2layers v3 was achieved. RvdE developed the original version of WAM2layers and performed the literature analysis of WAM2layers papers.

*Competing interests.* The authors declare no competing interests.

*Acknowledgements.* The authors acknowledge funding from the Netherlands eScience Center (Grant ID: 027.021.S07), the Netherlands Organization for Scientific Research (NWO) under the Open Science Fund (Grant ID: OSF23.1.029) and the Mainstreaming OS Fund from
TU Delft. We thank Yang Liu, Bart Schilperoort and Vincent de Feiter for also contributing significantly to the code for WAM2layers v3. Furthermore, we thank all other contributors to WAM2layers and its user community on GitHub.



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
