# Peer review of "Atmospheric moisture tracking with WAM2layers v3"

_EGUsphere, 2024_

## Referee Comment (RC2)

This article provides a detailed introduction to the atmospheric moisture tracking model WAM2layers v3, covering its history, model formulas, numerical implementation, application cases, best practices, limitations, and future development directions. It serves as a comprehensive record of the model and offers important reference for research in related fields. That said, I have some concerns for the model authors.

First, the model's basic algorithm is still the same as its last version with similar limitations, i.e., the Eularian grid computation introducing instability over the polar areas. As a result, the polar grids are to be excluded. This caused a big problem in its application as the polar region can't be accounted within. There are many solutions that came up by the GCMs developers as bringing in the Gaussian grid, or using irregular grids like triangulars. I wonder if the authors had ever tried these methods to solve this edge problem?

L344, … and exacerbates numerical diffusion.

Here, if I understand it right, using too small a time step will not do even when the computation capability allows, because it exacerbates numerical diffusion? If so, which range will be good for this numerical scheme? Like the Courant number should not exceed unity but larger than 0.5? Is it possible to use variable time steps across the longitude? As Fig. 5 shows, in the equator, the grid is larger, the time step can be large; over high latitudes, the grid is smaller, the time step becomes small?

L426, 'upstream' or 'Godunov' scheme is known to be numerically diffusive.

If this upstream scheme is numerically diffusive, why not choosing other suitable schemes? Such as Lax-Wendroff scheme?

L450, Generalizing to N layers could potentially simplify the code

How could that possible? For example, dividing into 3 layers can simplify the code more than the current two layers?

Fig. 6. 54% was associated with transport across the domain boundaries.

What is the domain of this case? How can more than 50% of the moisture be lost through boundaries?

L358, Similary to similarly

L376, Van der Ent to "van der Ent", and all the references as well from L720-730

---

## Author Response (AR1)

**Point-by-point response**

Referee comments in black; *responses in blue*; *manuscript changes in red (line numbers refer to the diff file)*.

**Reviewer 1**

Review of Atmospheric moisture tracking with WAM2layers v3 by Peter Kalverla , Imme Benedict, Chris Weijenborg, and Ruud J. van der Ent

This study describes version3 of the WAM2layers model which is a Eulerian offline moisture tracking model. WAM2layers v3 is rewritten in Python from previous model code and shared following FAIR principles. This manuscript describes the model code and setup in detail, discusses model limitations and potential future developments and provides two use cases.

The manuscript gives a detailed and well-structured overview of the model principles and how it is implemented in python. Further, the efforts to share the code in a user-friendly way and to reach out to the community are a good example of code sharing and development. While the new model version is a good role model of maintaining research software, the description of model limitations and potential shortcomings is sometimes only vaguely formulated. Further, the readability of the introduction should be improved and the use cases' outcomes better explained. As this study provides a substantial contribution to modelling science, and especially the moisture tracking community where previous versions of the model have been used in many studies, I recommend it for publication in GMD after addressing the following line-by-line comments:

*Dear reviewer, thank you for the comprehensive review! Your detailed comments provide valuable insight into how the manuscript comes across and highlight several areas where it can and should be improved. A more detailed response is included below with your own comments for easy reference.*

Figure 1: Can you add a representation of the tagged moisture to the conceptual figure? A visualisation of the different moisture fluxes and/or moisture budgets could help to follow the equations.

*You are right to highlight Figure 1, as it was mostly created as an appetizer, but could instead be used to help the reader understand different aspects of the model. To this end we propose to split the figure in two subfigures, one illustrating the domain configuration, the other zooming in on a single cell for the illustration of the different fluxes.*

*New figure 1 and references to it on L47, L123, L143, L217*

27-51: The introduction contains a long description of different types of moisture tracking models. This part is difficult to read and not very relevant for this study. As these types are also summarised in the referenced studies, the detailed discussion could be removed from this manuscript.

*Thanks for pointing this out. Our attempt to place WAM2layers in a broader context has escalated a bit. We agree that it makes sense to cut down the text.*

*Removed text between L26 - L42*

65-83: This list does not contain much information. Could this information be provided in a more informative way? E.g in a table, where the references are listed?

*The wide usage of WAM2layers is one of the key motivating factors for this manuscript, and we deem it important to mention previous use cases. That said, we do agree that putting this information in a table (perhaps next to figure 2) will prevent it from interrupting the flow of the main text.*

*Bullet list on L70 removed in and introduced table 1*

97: "This facilitates..." What does "this" refer to?

*It refers to the "generic description" of the governing equations. This was included to make it clear that the two-layer concept of WAM2layers is not immediately evident from these equations. We can reformulate this to make it clear that the two-layer concept emerges only later in the manuscript.*

*Reformulated on L101*

229-230: "This distributes cloud and rain/snow water across all levels, which is not perfect, but better than not counting it at all." Water is not distributed equally in the vertical column, with the majority residing in the lower troposphere. This is neglected if the cloud, rain and snow water is distributed equally across all levels. Does it make a difference for moisture tracking if the cloud and rain/snow water distribution is weighted by the relative water content of each layer? Further, "better than not counting it at all" is very vague. Can you be more precise in your statement?

*What you suggest here is in fact what is being done, but we understand that this was not clear from the description. A better phrasing would be "This proportionally adds missing cloud water to all levels." Moreover, in the preceding text and equation, we should distinguish more clearly here between the* calculated *column water* vapour *and the ERA5* total *column water. The correction factor thus obtained is applied to each level individually*

*(see multiplication factor $S_K$), which means it is weighted by the relative water content of that layer. We can consider writing $S_k$ as numerator and put the ERA5 factor outside the fraction*

*Reformulated text used on L243 and additional clarifications on L240 and eq 19.*

267: "WAM2layers v3.1" I think this is the first time that you mention v3.1. Is there a difference to v3?

*We use semantic versioning in our code, which consists of three numbers, vX.Y.Z, where X is a major, Y is minor, and Z is patch release. When we mention v3 (without specific minor/patch number), we refer to the major version only, i.e. that encompasses 3.1.0 and 3.2 etc.. Within a major version we strive for backwards compatible changes, such that the manuscript (and dependent code) will still be valid, even if new features are added. We will clarify this in the manuscript. In the introduction we already discuss previous versions, so it makes sense to add it there.*

*New paragraph on semantic versioning L60*

282: "the data is first interpolated to a finer time step" What is the new temporal resolution with "a finer time step"?

*This is configurable and may depend on the domain of interest. More details are discussed in section 3.4. However, we could mention a rough number here and point to section 3.4 such that readers will not have this question stuck in their minds.*

*Change on L299.*

291: " Sijt represents the total column water" But not Qtc, correct? Can you make it clearer that this is a "different" total column water than Qtc?

*Correct, we agree this is confusing. We could change the description here to something like "represents the calculated column total, whether or not corrected for cloud/rain/snow liquid water as per eq (19).". With respect to the naming of Qtc: In preparing this manuscript we switched from Q to S to denote the grid-cell integrated water vapour. Following this rationale it makes sense to rename the ERA5 total column water as well, but it seems we forgot that one. In response to comment on line 229 we've renamed it to S_T_ERA5. The updated formulation there should also help to clarify the point raised here.*

*Reformulated on L209*

318: " the vertical transport terms are directed from the upper to the lower layer, i.e., positive downward" This statement is repeated many times. Can you check if it is really needed each time?

*"Positive downward" is mentioned three times, but in different circumstances. The first occurrence relates to a note about the direction of precipitation and evaporation in ERA5; the second to the resolution of the vertical flux in the general case of n layers; the last to the specific equation in our two-layer model.*

*In principle, the third is implied by the second and could be omitted. However, we have seen several misunderstandings about this in the past, and we ourselves have also been quite confused at times. Therefore, we see value in iterating this point. That said, we will omit the "i.e. positive downward" from the quoted sentence, and reformulate the rest to stress our focus, e.g. "the only remaining vertical transport term, which is shared by both layers".*

Omitted on L337; added note on shared term om L336.

358: "boundary of the domain": It was not immediately clear to me, which domain this is – model domain or the tagging domain? Further, what do you consider as a significant boundary transport (compare also comment on Fig. 6)?

*The term domain is used for the entire area over which WAM2layers is applied. In this case this is the area shown in Fig.6: [50 W, 30 N, 60 N, 30 W]. The definitions of 'domain' and 'tagging region' will be clarified in an updated Fig. 1. The follow-up question is answered in the response to the comment on Fig. 6 (i.e., next comment).*

New figure 1 and references to it on L47, L217

Figure 6: "By then, 42 % of moisture was tracked to its source; 3.4 % of was still in the domain's atmosphere, and 54 % was associated with transport across the domain boundaries." 42% tracked to the sources seems like a low number. Does this use case represent the typical performance of WAM2layers, and is this the expected tracking efficiency? Having 54% of moisture that is transported across the domain boundaries, what do you consider a significant transport across the boundaries (compare lines 358-359)?

*The amount of moisture transported across a domain is not per se a good indicator for 'the performance of WAM2layers', but rather related to the choice of the domain size and associated research questions. It is true that some moisture tracking models estimate moisture contributions in general to be more local or more remote, but we consider that discussion out of scope here. If one would like to get an indication of the main sources, 42% precipitation tracked back surface evaporation could be sufficient as the 54% of moisture that is transported across the domain boundaries likely constitutes a large combination of many small sources (<0.1 mm). If one has the study goal to identify, for*

*instance, 80% of the moisture sources, one could conclude that the domain is too small and that the case should be rerun with a larger domain. Long story short, 'significant' is case-dependent. We will rephrase to make this clearer.*

*Clarified domain boundary concept in Figure 1 and references to it; added note on domain boundary interpretation on L378; added note on limited domain on L396.*

372-374: "The spatial pattern of the sources corresponds to the sources determined by Insua-Costa et al. (2022) and Staal and Koren (2023)." How do the WAM2layer sources compare in terms of tracked moisture? You mention that a model intercomparison study is currently done, and I acknowledge that such a question can be investigated in more detail in an intercomparison study. But as the high loss/diffusion (?) of moisture is striking in this use case, this question comes immediately to my mind when you compare the WAM2layers patterns to other studies.

*Insua-Costa et al. (2022) and Staal and Koren (2023) also estimated significant remote contributions:*

| Region | Insua-Costa et al. (2022) | Staal and Koren (2023) |
|---|---|---|
| Europe and North Africa | 51% | 50% |
| North Atlantic | 11% | 24% |
| Mediterranean and Black Sea | 6% | 3% |
| North and Baltic Sea | 3% | 3% |
| Tropics | 15% | 3% |
| North America | 10% | 14% |
| Other remote | 4% | 3% |

*Our domain excludes the regions 'Tropics', 'North America' and 'Other remote' and does not fully cover the other regions either, because we miss Scandinavia, Eastern Europe until 50 E and the Western part of the North Atlantic. The equivalent boundary transport if they would have had the same domain in the cases of Insua-Costa et al. (2022) and Staal and Koren (2023) would have been >>29% and >>20%, respectively. Without having a more detailed comparison we do not think it should be concluded that WAM2layers is necessarily much more diffusive than other moisture tracking models and leave such a comparison to the more comprehensive intercomparison study that is underway. We will add a short discussion in the revised manuscript without going into too much detail.*

*Short discussion added on L396*

Figure 7: "Of the tagged moisture 7.8 % recycled within the source region." How is recycling defined in this forward tracking mode?

*This follows the definition of regional evaporation recycling (van der Ent and Savenije, 2011), but for clarity we will change this to: "of the tagged evaporation 7.8 % precipitated within the source region."*

*Changed in caption of Figure 7*

465: What are "quick" regional moisture recycling calculations?

*Understandably this is not immediately clear, however, we also did not want to add too much detail here, because the feature is currently not incorporated. With the current version one can also do single grid cell regional moisture recycling calculations, but it would require to specify a new tagging region for each tracking experiment. If, however, one makes the assumption that tracked moisture that leaves a grid cell never returns, it is possible to calculate the recycling within each grid cell in a single experiment, making the computations much quicker. We propose to change the sentence to: "such calculations for moisture recycling within a single grid cell for all grid cells of the domain at once (van der Ent and Savenije, 2011; De Hertog et al, 2024)", which is hopefully more descriptive, but leaves the details in the references nonetheless.*

*Changed on L505*

Minor comments:

43: "or something in  between": Can you be more precise?

*Outdated; after a comment on this paragraph that line has been removed.*

*L38*

161: "…starting isobaric coordinates to yields…" something is missing here

*Remnant of old text, will be removed*

*L170*

164 " the horizontal transport in terms in (2) can be written" delete in terms

*Thanks for spotting*

*L173*

204: water → humidity?

*Better indeed. Too bad this breaks the alliteration…*

*L213*

242: "mask of values between 0 and 1" Do you mean mask of values of 0 or 1?

*No, we mean between. A value of 0.5 could be useful e.g. at country borders, where half the grid cell is inside the domain of interest. We will clarify that.*

*Decided to omit that implementation detail as it is explained in the user guide and not super relevant for the manuscript. L256.*

302: "where subscript t denotes the column totals" This should be introduces when t is used for the first time.

*Thanks for pointing us to the inconsistent use. We will use capital T here and also in the total column water correction (currently tc). That would more clearly differentiate this subscript with the superscript t for time.*

*L309; L320; eq 23, 24, 25; L240, eq 19*

349: remove "a such that"

*Thanks*

*L368*

350: "These limits are not 100 % watertight" Colloquial expression

*We will reformulate that*

*L369*

369: "the, Ourthe" remove comma

*Thanks*

*L390*

379: "similar moisture source patterns" do you mean "moisture sink patterns"?

*Yes, thank you.*

*L403*

402: " WAM2layers ships with " Colloquial expression

*"Includes" is indeed better*

*L423*

430: "get a better grip on it" Colloquial expression → it is not clear what you exactly mean with "a better grip"

*We're not sure either.. We will reformulate that to put more emphasis on trying to quantify numerical diffusion.*

*L467*

435: " we should be careful not to throw the baby out with the bathwater." Colloquial expression → can you spell this out, or reconsider if this statement is needed?

*We will reformulate this*

*L473*

**Reviewer 2**

This article provides a detailed introduction to the atmospheric moisture tracking model WAM2layers v3, covering its history, model formulas, numerical implementation, application cases, best practices, limitations, and future development directions. It serves as a comprehensive record of the model and offers important reference for research in related fields. That said, I have some concerns for the model authors.

*We sincerely thank the reviewer for their thoughtful feedback. It's clear they picked up on some of the numerical limitations of WAM2layers, which is something we deliberately highlighted in the manuscript. In that sense, their comments are exactly the kind of discussion we hoped to encourage.*

*With this manuscript, we have two main goals. First, we want to provide a reference for researchers using WAM2layers, helping them interpret past results while also being aware of its limitations – especially since we have seen cases where the model was used inappropriately. Making fundamental changes to the algorithm would make it difficult to serve this purpose. Second, we want to lay a solid foundation for future improvements, rather than trying to solve all existing limitations at once, which would go beyond the scope of this work. As such, we see the improved code structure in WAM2layers V3 and the updated documentation in the form of this manuscript as a starting point for further development. The reviewer's suggestions will help to enhance the manuscript and, moreover, they inform future developments as well.*

First, the model's basic algorithm is still the same as its last version with similar limitations, i.e., the Eularian grid computation introducing instability over the polar areas. As a result, the polar grids are to be excluded. This caused a big problem in its application as the polar region can't be accounted within. There are many solutions that came up by the GCMs developers as bringing in the Gaussian grid, or using irregular grids like triangulars. I wonder if the authors had ever tried these methods to solve this edge problem?

*The brief answer is that we have not, as our research has focused more on the tropics and midlatitudes, for which the current implementation suffices. On the other hand, we have seen (unpublished) cases where WAM2layers was applied in these regions, typically not because the research question required it, but because the users did not recognize this limitation. This is why we highlighted it in the manuscript.*

*It is interesting, though, to briefly consider how we could facilitate such use cases rather than warn against them. Historically, WAM2layers has always operated on the same grid as*

*the input data, i.e. reanalysis and climate model output that is typically retrieved at an equirectangular latitude-longitude projection. Assumptions about this grid structure are deeply woven into the existing implementation, and perhaps the easiest way to accommodate moisture tracking over polar areas is to start again from scratch. Until now we have resisted this temptation, as we also foster continuity of the codebase, especially to keep existing users onboard and avoid alienation. Following a more gradual path instead, WAM2layers V3 includes substantial efforts to improve the code structure, isolating the solver and related numerical aspects from other parts of the code that deal with things like I/O and orchestration. This enables experimentation with different solvers, although it would probably make sense to start with solvers that operate on the same grid.*

*As a workaround, for research questions that focus on moisture transport over polar areas (but not globally), it might be possible to remap the input data to another equirectangular grid that is optimized for polar areas. This would keep the existing assumptions intact, but it would still require substantial code changes that are beyond the scope of this manuscript.*

*We thank the reviewer for inviting us to ponder over this edge case, and suggest to include this discussion in the revised manuscript.*

*A redacted version of this response is included in the manuscript on L449, i.e. the first discussion point, emphasizing its importance. We also added specific notes on input data structure and a forward reference to this discussion on section 4.3 (L358, L361).*

L344, ... and exacerbates numerical diffusion. Here, if I understand it right, using too small a time step will not do even when the computation capability allows, because it exacerbates numerical diffusion? If so, which range will be good for this numerical scheme? Like the Courant number shouldnot exceed unity but larger than 0.5? Is it possible to use variable time steps across the longitude? As Fig. 5 shows, in the equator, the grid is larger, the time step can be large; over high latitudes, the grid is smaller, the time step becomes small?

*This interpretation is correct on all accounts, and the question about variable time steps is interesting. Ideally, rather than making it latitude-dependent, we could calculate the local Courant number on the fly and use that to inform the time step. What complicates this is that the model requires matching input data for all time steps and all grid cells. We'd need to find a way to do that efficiently, e.g. by assuming that the total moisture flux field is constant between input times. Our issue tracker has an open ticket that suggests calculating the CFL during preprocessing and suggesting a time step based on that. We have added the suggestion to use a latitude-dependent time step to that ticket. Implementing this is beyond scope for the current manuscript.*

L426, 'upstream' or 'Godunov' scheme is known to be numerically diffusive. If this upstream scheme is numerically diffusive, why not choosing other suitable schemes? Such as Lax-Wendroff scheme?

*One favourable property of the Godunov scheme is that it doesn't permit overshoot or undershoot. Intuitively, this makes a lot of sense for a moisture tracking model, considering the absolute minimum of 0 (tagged) moisture.*

*Interestingly, the textbook by Cushman-Roisin and Beckers, which we've consulted heavily during the preparation of this manuscript, also discuss the Lax-Wendroff scheme as a higher-order alternative, but they immediately show that this comes at the cost of introducing dispersion and over/undershoot. They proceed to introduce predictor-corrector methods and flux-limiter schemes. During WAM2layers' early development we were mostly oblivious to some of the more advanced options available in the atmospheric modelling community, and it is fair to say that we are now only trying to catch up. For WAM2layers V3, we have made substantial efforts to isolate the solver, exactly to facilitate gradual experimentation with alternatives like the ones mentioned above. However, we see this as future work where the current manuscript can serve as a point of reference.*

*We have tried to explain this in the manuscript, but from the comment it is clear that we should explain this more clearly.*

*We have rewritten the paragraph (L460) such that it starts by emphasizing the strong points, especially that it prevents over/undershoot (that was not explicitly written before). We retained the references to some proposed alternatives, which we believe is in line with the reviewer's recommendation. As it is now, we believe the paragraph more clearly emphasizes that WAM2layers v3 makes a substantial effort towards using alternative solvers.*

L450, Generalizing to N layers could potentially simplify the code How could that possible? For example, dividing into 3 layers can simplify the code more than the current two layers?

*The idea here is that we could skip the preprocessing step and operate directly on the input data. Most of section 3 would then become obsolete, and the governing equations could be formulated directly in terms of wind and humidity rather than summed fluxes (eq 16) that we currently use. It is unlikely that this will work well with only 3 layers, but since we are already processing input data on e.g. 20 layers, why not try to work with those levels directly in the tracking routine? As mentioned in the text though, it remains to be seen whether this improves the results.*

*Added a note on L489 that these N layers should correspond to the layers in the input data*

Fig. 6. 54% was associated with transport across the domain boundaries. What is the domain of this case? How can more than 50% of the moisture be lost through boundaries?

*The domain in this case this is the area shown in Fig.6: [50 W, 30 N, 60 N, 30 W]. The definitions of 'domain' and 'tagging region' will be clarified in an updated Fig. 1. The amount of moisture transported across a domain is related to the choice of the domain size and associated research questions. The 54% of moisture that is transported across the domain boundaries, in this case, constitutes a large combination of many small sources (<0.1 mm). If one has the study goal to identify, for instance, 80% of the moisture sources, one could conclude that the domain is too small and that the case should be rerun with a larger domain. In this case, another reason to limit the spatial extent of the domain is that we want to provide first-time users with an example case that is quite lightweight in terms of data and compute requirements. We will rephrase to make this clearer.*

*Clarified domain boundary concept in Figure 1 and references to it; added note on domain boundary interpretation on L378; added note on limited domain on L396.*

L358, Similary to similarly

*Thanks for spotting!*

*L376*

L376, Van der Ent to "van der Ent", and all the references as well from L720-730

*Consider it done.*

*L687 – L705 in new manuscript (diff file was generated with old references, and creating a new diff will mess up all line numbers quoted in this line-by-line response).*

---

## Author Response (AR2)

Three minor corrections from reviewer have been applied